**Data Availability Statement:** All relevant data are within the paper and its S1 Appendix files.

**Funding:** This study was jointly supported by grants from the Natural Science foundation of

# Effects of salt stress on the photosynthetic physiology and mineral ion absorption and distribution in white willow (*Salix alba* L.)

Xin Ran[1], Xiao Wang[1], Xiaokuan Gao[2], Haiyong Liang[1], Bingxiang Liu[1,3]*, Xiaoxi Huang[1]

1 College of Forestry, Hebei Agricultural University, Baoding, Hebei, China, 2 College of Life Science, Hengshui University, Hengshui, Hebei, China, 3 Hebei Urban Forest Health Technology Innovation Center, Baoding, Hebei, China

* proser211@126.com

## Abstract

### Objective

The purpose of this study was to explore the adaptive mechanism underlying the photosynthetic characteristics and the ion absorption and distribution of white willow (*Salix alba* L.) in a salt stress environment in cutting seedlings. The results lay a foundation for further understanding the distribution of sodium chloride and its effect on the photosynthetic system.

### Method

A salt stress environment was simulated in a hydroponics system with different NaCl concentrations in one-year-old *Salix alba* L.branches as the test materials. Their growth, ion absorption, transport and distribution in the roots and leaves, and the changes in the photosynthetic fluorescence parameters were studied after 20 days under hydroponics.

### Results

The results show that The germination and elongation of roots are promoted in the presence of 171mM NaCl, but root growth is comprehensively inhibited under increasing salt stress. Under salt stress, $Na^+$ accumulates significantly in the roots and leaves, and the $Na^+$ content and the $Na^+/K^+$ and $Na^+/Ca^{2+}$ root ratios are significantly greater than those in the leaves. When the NaCl concentration is $\leq$ 342mM, *Salix alba* can maintain relatively stable $K^+$ and $Ca^{2+}$ contents in its leaves by improving the selective absorption and accumulation of $K^+$ and $Ca^{2+}$ and adjusting the transport capacity of mineral ions to aboveground parts, while $K^+$ and $Ca^{2+}$ levels are clearly decreased under high salt stress. With increasing salt concentrations, the net photosynthetic rate ($P_n$), transpiration rate (E) and stomatal conductance ($g_s$) of leaves decrease gradually overall, and the intercellular $CO_2$ concentration ($C_i$) first decreases and then increases. When the NaCl concentration is < 342mM, the decrease in leaf $P_n$ is primarily restricted by the stomata. When the NaCl concentration is > 342mM, the decrease in the $P_n$ is largely inhibited by non-stomatal factors. Due to the salt stress environment, the OJIP curve (Rapid chlorophyll fluorescence) of *Salix alba* turns into an

Hebei Province(17226320D-4). The funders had no role in study design, data collection and analysis, decision to publish, or preparation of the manuscript.

**Competing interests:** The authors have declared that no competing interests exist.

OKJIP curve. When the NaCl concentration is > 171mM, the fluorescence values of points I and P decrease significantly, which is accompanied by a clear inflection point (K). The quantum yield and energy distribution ratio of the PS reaction center change significantly (φPo, Ψo and φEo show an overall downward trend while φDo is promoted). The performance index and driving force ($PI_{ABS}$, $PI_{CSm}$ and $DF_{CSm}$) decrease significantly when the NaCl concentration is > 171mM, indicating that salt stress causes a partial inactivation of the PSII reaction center, and the functions of the donor side and the recipient side are damaged.

## Conclusion

The above results indicate that *Salix alba* can respond to salt stress by intercepting $Na^+$ in the roots, improving the selective absorption of $K^+$ and $Ca^{2+}$ and the transport capacity to the above ground parts of the plant, and increasing φDo, thus shows an ability to self-regulate and adapt.

## 1 Introduction

As one of abiotic stresses, salt stress can significantly affect plant growth and yield. Today, 1.125 billion hectares of farmland around the world are threatened by salt stress, which is an important issue for agriculture [1]. China also has 367 billion hectares affected by salt stress, accounting for 1,230 hectares of farmland soil [2]. Among the areas of concern, the coastal area, one of the primary types of saline-alkali land, has frequent water-salt interactions and secondary salinization as it is close to the sea [3]. Because the ecological environment of the salinization area is fragile and natural conditions are limited by many factors, it is highly significant to develop and use saline-alkali land scientifically and rationally while under pressure from a rapid population increase and a sharp decline in land resources; the goal is to advance towards the sustainable and healthy development of China's forestry and ecological environment [4].

Choosing and cultivating excellent salt-tolerant tree species through biotechnology is currently one of the most economical, effective, ecological and environment-friendly biological measures to solve the soil salinization problem [5]. *Salix babylonica* is a deciduous tree or shrub belonging to *Salix* in the family *Salicaceae*. It has strong ecological adaptability and can grow well under saline-alkali, drought and barren soil conditions [6]. Previous studies on the salt tolerance of *Salix* plants were mostly focused on the physiological responses of seedlings to salt stress [7–9], but there are few studies on the adaptability of plants to salt stress specific to seedlings. High salt stress will cause plant water loss, ion imbalance and nutrient element deficiency through osmotic stress and ion poisoning [10], which will affect the normal growth and morphology of plants. A series of physiological growth changes in plants under salt stress are the comprehensive embodiment of their salt tolerance ability, among which the growth status of plant roots, the ion accumulation in different organs and the change in photosynthetic fluorescence parameters are important factors affecting the salt tolerance ability of plants [11–13]. These indicators not only can represent the extent of the effects of stress factors on plants, but they can also reflect the growth of plants under salt stress, the selective absorption and transport of ions, and the photosynthesis ability. Willow has the characteristics of antipyretic, analgesic, anti-inflammatory, anti-rheumatism, astringent, drought resistance [14] and anti-corrosion [15], among which the bark of White Willow contains salicin [16, 17] with

antibacterial, bactericidal, antioxidant, antipyretic, analgesic and other functions, and is a good natural food additive and food resource of health care products [18]. Its roots can also absorb harmful elements, reduce the impact of harmful elements on the surrounding soil [19, 20], and play a role in purifying polluted water [21]. *Salix alba* has strong adaptability [22], so it has great potential for use and promotion in the ecological management of coastal saline-alkali soil. Therefore, this experiment involved *Salix alba* branches as the object and used hydroponics to simulate the seedling raising process of cuttings on coastal saline-alkali land to study the growth of their roots, the changing ion contents in the roots and leaves, and the changing photosynthetic fluorescence parameters under different salt concentrations. Exploring the salt tolerance of *Salix alba* can not only provide theoretical reference for the study of its salt stress adaptation mechanism, but also provide suggestions for the utilization of salix salix salix in coastal saline-alkali areas.

## 2 Materials and methods

### 2.1 Test materials and test design

The test materials were collected from the germplasm resource nursery of Golden Beach Forest Farm in Huai'an County, Hebei Province. The branches of the *Salix alba* were basically the same in terms of growth, and those that were robust and free of diseases and insect pests were selected after the leaves fell in December. The middle two-thirds of the selected branches were cut into 20 cm-long cuttings. The uppermost bud was 0.5–1 cm from the top of the cuttings. The upper cut was a flat cut, and the lower cut was an oblique cut. The experiment was performed in the Artificial Climate Room of Hebei Agricultural University, Baoding City, Hebei Province on December 23, 2019. The temperature of the climate room was set to 28˚C/25˚C (light/dark); the LED cold light source maintained the light intensity at 1000 $\mu mol\cdot m^{-2}\cdot s^{-1}$; the photoperiod was 14 h/10 h (light/dark); and the humidity was 60%.

The test material was placed in a 55 cm×38 cm×15 cm (length ×width ×height) plastic box for hydroponic culture (Fig 1). The experiment consisted of 5 treatments, and each treatment was repeated 3 times. We set the concentration of Nacl and the composition of culture medium by referring to existing studies [23–25]. A 1/2 dilution of Hoagland's complete nutrient solution was used as the base to prepare hydroponic solutions with NaCl concentrations of 171mmol342mmol513mmol684mmol, and 1/2 Hoagland's complete nutrient solution (PH = 7.2) was used as a control (CK). 1/2 Hoagland's complete nutrient solution includes: Ca $(NO_3)_2\cdot 4H_2O$ 472.5mg$K_2SO_4$ 303.5mg$NH_4H_2PO_4$ 57.5mg$MgSO_4$ 246.5mg$NaFeC_{10}H_{12-}N_2O_8\cdot 3H_2O$ 30mg$FeSO_4$ 15mg$H_3BO_3$ 2.86mg$Na_2B_4O_7\cdot 10H_2O$ 4.5mg$MnSO_4$ 2.13mg, $CuSO_4$ 0.05mg$ZnSO_4$ 0.22mg$H_8MoN_2O_4$ 0.02mg. There were 25 cuttings in each treatment, and they were soaked directly in the solution; the height of the solution was approximately more than half of the height of the cuttings. The nutrient solution was changed every 5 days during the growth process. Before the nutrient solution was changed, the cuttings were removed and the roots were rinsed with water to wash away the last residual salt and prevent excessive salt accumulation. The contents of Na$^+$, K$^+$ and Ca$^{2+}$ ions and the photosynthetic parameters and chlorophyll fluorescence kinetic curve parameters in the roots and leaves were determined following 20 days of treatment.

### 2.2 Measured items and methods

**2.2.1 Measurement of root growth parameters.**   During the growth of *Salix alba*, the number of root sprouting days and the rooting rate of all the cuttings were counted, and the rooting index was calculated according to the number of root sprouting days (rooting index = ∑Gt/Dt, where Dt, the day of the rooting test; Gt, the number of rooting branches on the day,

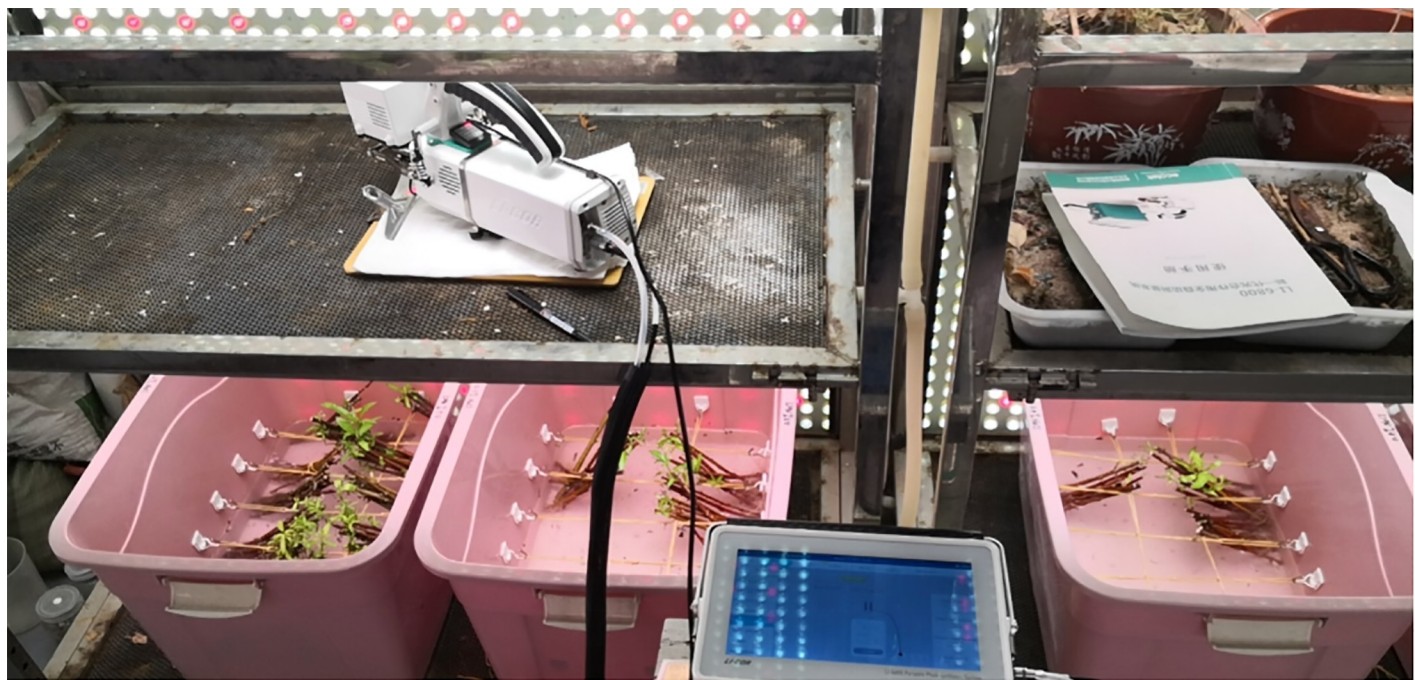

**Fig 1. Experimental scenes of the effects of salt stress on *Salix alba* L.**

and the rooting index is the number of rooting branches on the day/sum of days). After 20 days of salt stress treatment, 5 uniformly growing cuttings were selected to measure the average root number and average root length.

**2.2.2 Determination of ion contents in the roots and leaves, calculation of the ion selective absorption and transport ratio.** The measurement method for tracking the ion content was slightly modified relative to the method by Yang Sheng et al. [26] and Yu Bingjun et al. [27]. The sample was first baked at 105˚C for 30 min and then dried at 70–80˚C to a constant weight. After the sample was ground and passed through a sieve (the aperture was 0.425 mm), the fixed mass was weighed. Thirty mL of deionized water was added to the sample, which was then shaken well and placed in a boiling water bath for 2 h. After cooling, the sample was filtered and diluted to 50 mL. The $Na^+$, $K^+$ and $Ca^{2+}$ contents were determined by atomic absorption method (Atomic absorption spectrometer: ZEEnit700-700P; analytikjena computer in Germany). The methods of Zheng Qingsong et al. [28] and Yang Xiaoying et al. [29] were used to calculate the selective absorption and transport coefficients of ions X ($K^+$ and $Ca^{2+}$) by the roots and leaves according to the following formula. Ion absorption coefficient $SA_{x, Na}$ = root ($[X]/[Na^+]$)/medium ($[X]/[Na^+]$); ion transport coefficient $ST_{x, Na}$ = leaf ($[X]/[Na^+]$)/ Root ($[X]/[Na^+]$). In the formula, the $K^+$ content in the medium (culture broth) was 272 mg, and the $Ca^{2+}$ content was 230 mg.

**2.2.3 Determination of photosynthetic parameters in the leaves.** Following 20 days of salt stress treatment, the photosynthetic gas exchange parameters of the *Salix alba* leaves were measured. Five uniformly growing cuttings were selected for each treatment group in the test. After the cuttings were left under normal illumination in the climate room for 3 hours, we selected the 3rd to 5th leaves from the top to bottom with the same position, size, and light-receiving direction and with fully expanded functional leaves. Using a Li-6800 portable photosynthesis meter (LI-COR, USA), the $P_n$, E, $g_s$ and $C_i$ can be determined. The measurement

conditions were as follows: the PAR was 1000 μmol·m$^{-2}$ ·s$^{-1}$, the $CO_2$ concentration in the fixed system was 400 μmol·mol$^{-1}$, and the relative humidity was 60%.

**2.2.4 Rapid determination of the chlorophyll fluorescence induction kinetic curve.** After 20 days of salt stress treatment, 5 *Salix alba* cuttings with average growth were selected from each treatment for measurement. Before the measurement, the leaves were dark-adapted for 15 minutes, and then the rapid chlorophyll fluorescence induction kinetic curve and related parameters were measured using a Pocket PEA plant efficiency analyzer (Pocket PEA, Hansatech, UK). The resulting O-K-J-I-P curve was used for rapid chlorophyll fluorescence induction curve data analysis (JIP-test) and calculation [30, 31].

## 2.3 Data processing

One-way ANOVA and the LSD method were used to test the significance of the differences (α = 0. 05).

## 3 Results

### 3.1 Effects of salt stress on *Salix albicans* root growth

The test results show that although plants under salt stress can reach a 100% rooting rate between treatments, the average root number, average root length and rooting index are quite different among the treatments, and the overall trend is basically the same. The trend is that low-salt stress stimulates root germination and elongation, high-salt stress inhibits root growth, and the intensity of the inhibition is positively correlated with the salt concentration.

Figs 2 and 3 show that when the NaCl concentration was 171mM, the average root number and average root length were significantly increased compared with those of the control. This result may be a stimulating effect of low-salt stress on root growth and then appear again as the stress intensifies, with a gradual downward trend. When the NaCl concentration was 513mM, the root number and length were significantly lower than those of their respective controls by 48.7% and 39.9%, and the root growth was significantly inhibited at that time. Compared with the control, the rooting index did not change significantly when the NaCl concentration was 171mM, but with the increase in stress, the number of days for root

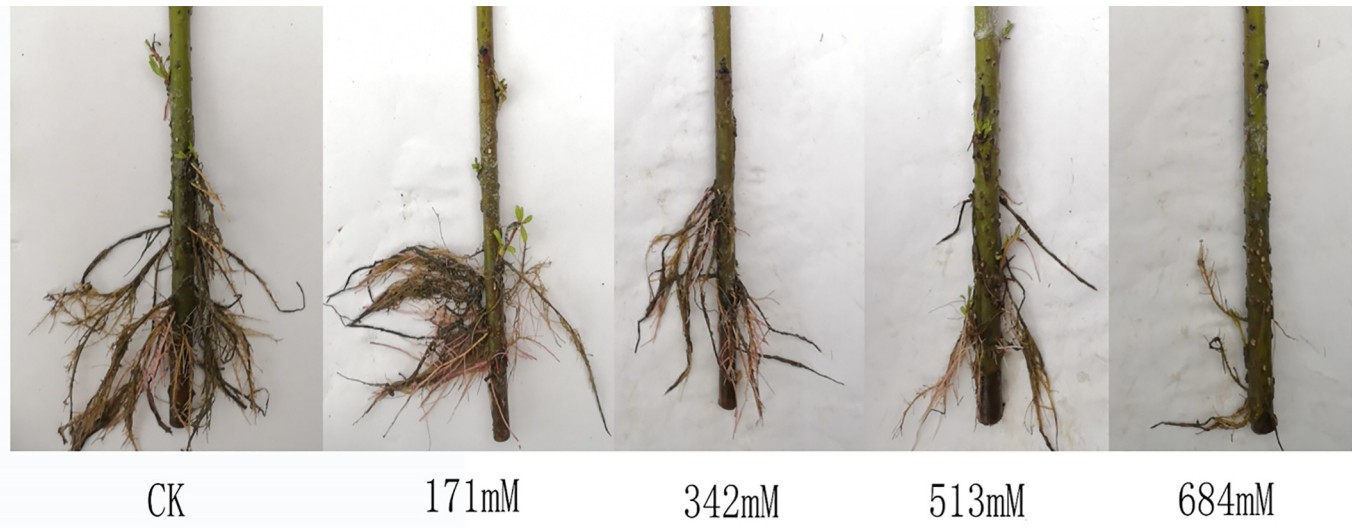

**Fig 2. Changes of root growth of *Salix alba* L. under salt stress.**

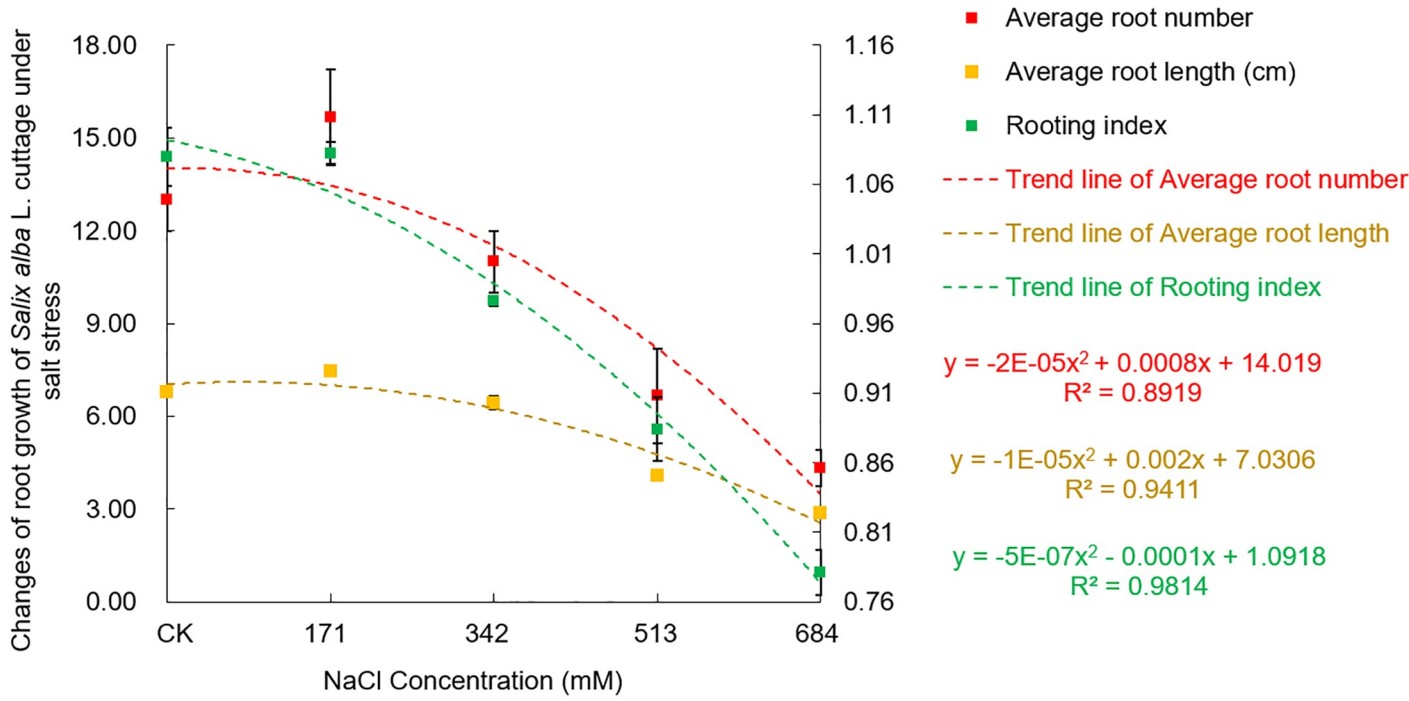

**Fig 3. Effects of salt stress on root growth of *Salix alba* L. cuttings.**

germination was delayed, and the rooting index decreased significantly. When the NaCl concentrations were 342mM, 513mM and 684mM, the rooting indexes were significantly lower than that of the control (9.6%, 18.1% and 27.7%).

## 3.2 Effects of salt stress on ion content, absorption and transport in the roots and leaves of *Salix alba*

The ion content measurements (Fig 4) showed that under different concentrations of NaCl, the $Na^+$ contents in the roots and leaves of *Salix alba* were significantly higher than that in the control group, and the range of $Na^+$ change was positively correlated with the stress concentration. The comparison of $Na^+$ contents in the roots and leaves shows that the $Na^+$ content of the roots is much higher than that in the leaves. Under 684mM NaCl stress, the $Na^+$ content in the roots could reach twice that in the leaves. With increasing stress concentration, the $K^+$ content in the leaves first increased and then decreased, reaching a peak at a concentration of 171mM NaCl, which was a significant increase of 14.0% compared to the control group. However, after the NaCl concentration was greater than 342mM, the concentration was significantly lower than that of the control. As the stress concentration increased, the $K^+$ contents in the roots of each treatment group showed a gradual decrease, which were all significantly lower than that of the control. The $Ca^{2+}$ content in the leaves of *Salix alba* increased first and then decreased with increasing salt concentration. At 342mM NaCl, compared with the control group, the concentration significantly increased by 13.6% and then showed a significant downward trend. The $Ca^{2+}$ content in the roots decreased continuously with increasing stress, and when the NaCl concentration was 684mM, the $Ca^{2+}$ content dropped to 35.6% of the control.

Figs 5 and 6 show that both the $Na^+/K^+$ and $Na^+/Ca^{2+}$ in the roots and leaves increased significantly with increasing NaCl stress concentration. This finding shows that as the stress

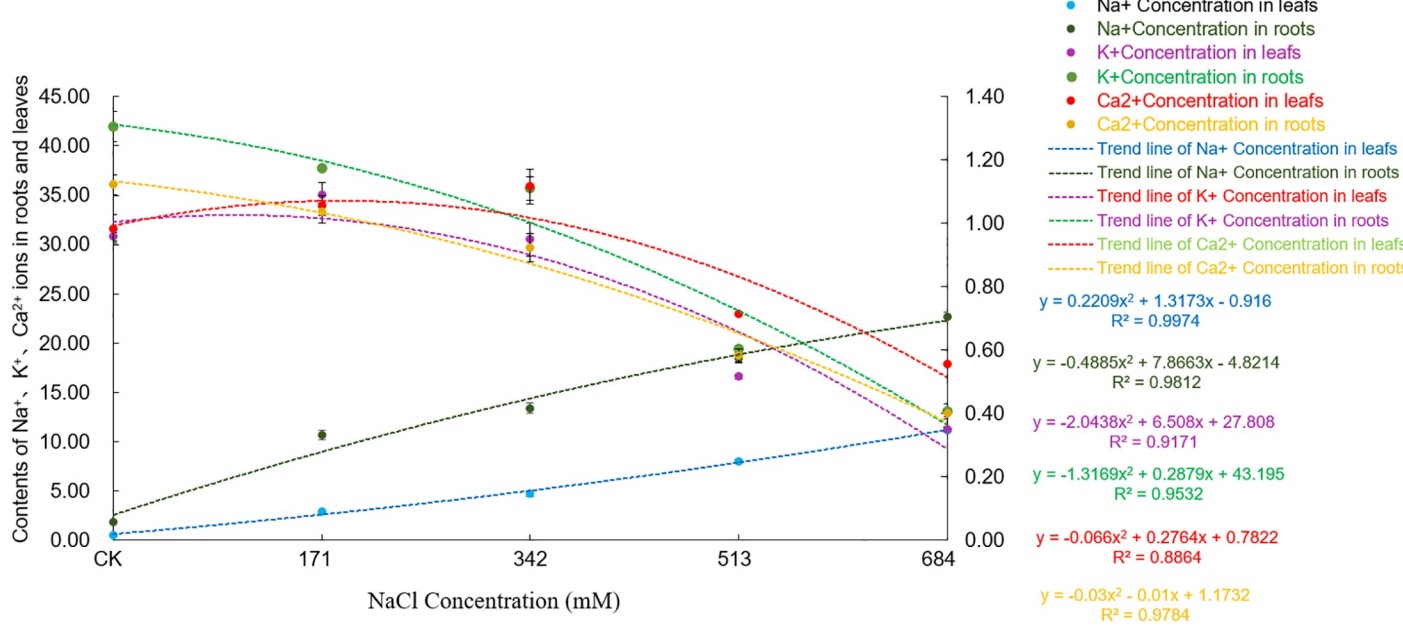

**Fig 4. Effects of salt stress on ion content in roots and leaves of *Salix alba* L.**

intensifies, the relative absorption of $Na^+$ by *Salix alba* increases greatly, but the absorption of $K^+$ and $Ca^{2+}$ decreases. The $Na^+/K^+$ and $Na^+/Ca^{2+}$ contents of all the treatments gradually decreased from root to leaf, and the rising $Na^+/K^+$ and $Na^+/Ca^{2+}$ in the roots were significantly (F = 1263.766, df = 4, Sig.<0.001; F = 10485.256, df = 4, Sig.<0.001) higher than those in leaves (F = 1235.223, df = 4, Sig.<0.001; F = 2335.783, df = 4, Sig.<0.001), suggesting that *Salix alba* could reduce the salt stress damage to young tissues by regulating ion transport.

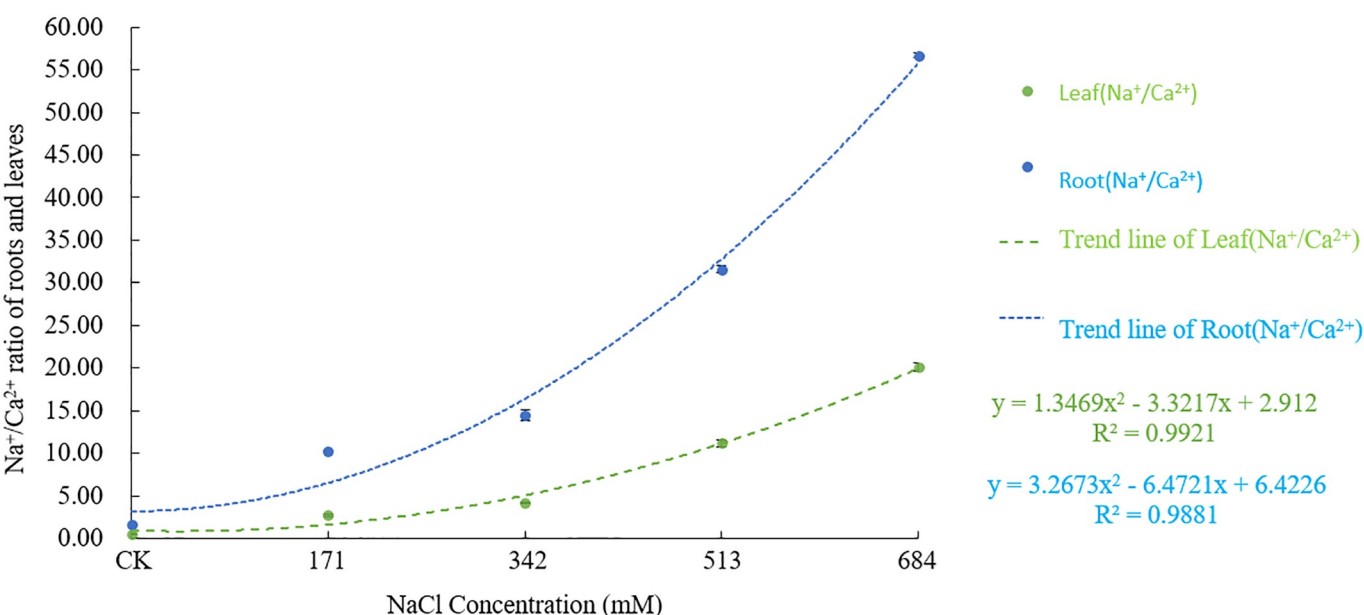

**Fig 5. Effects of salt stress on $Na^+/Ca^{2+}$ in roots and leaves of *Salix alba* L.**

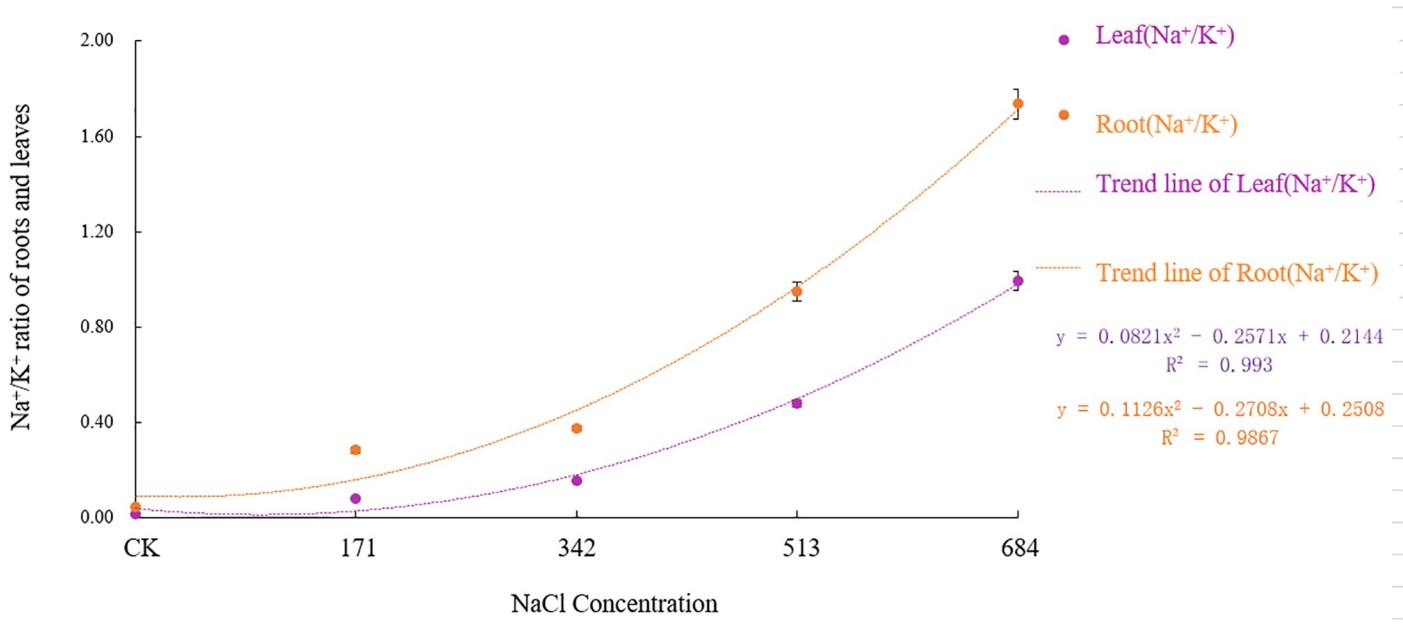

**Fig 6. Effects of salt stress on Na⁺/ K⁺ in roots and leaves of *Salix alba* L.**

As shown in Fig 7, with increasing NaCl stress, the $SA_{k, Na}$, $ST_{k, Na}$, $SA_{Ca, Na}$, and $ST_{Ca, Na}$ all showed a trend of first increasing and then decreasing. When the NaCl concentration was less than or equal to 342mM, the selective absorption capacity of the roots for $K^+$ and $Ca^{2+}$ and the selective transport capacity of the leaves for $K^+$ () and $Ca^{2+}$ were enhanced and reached a

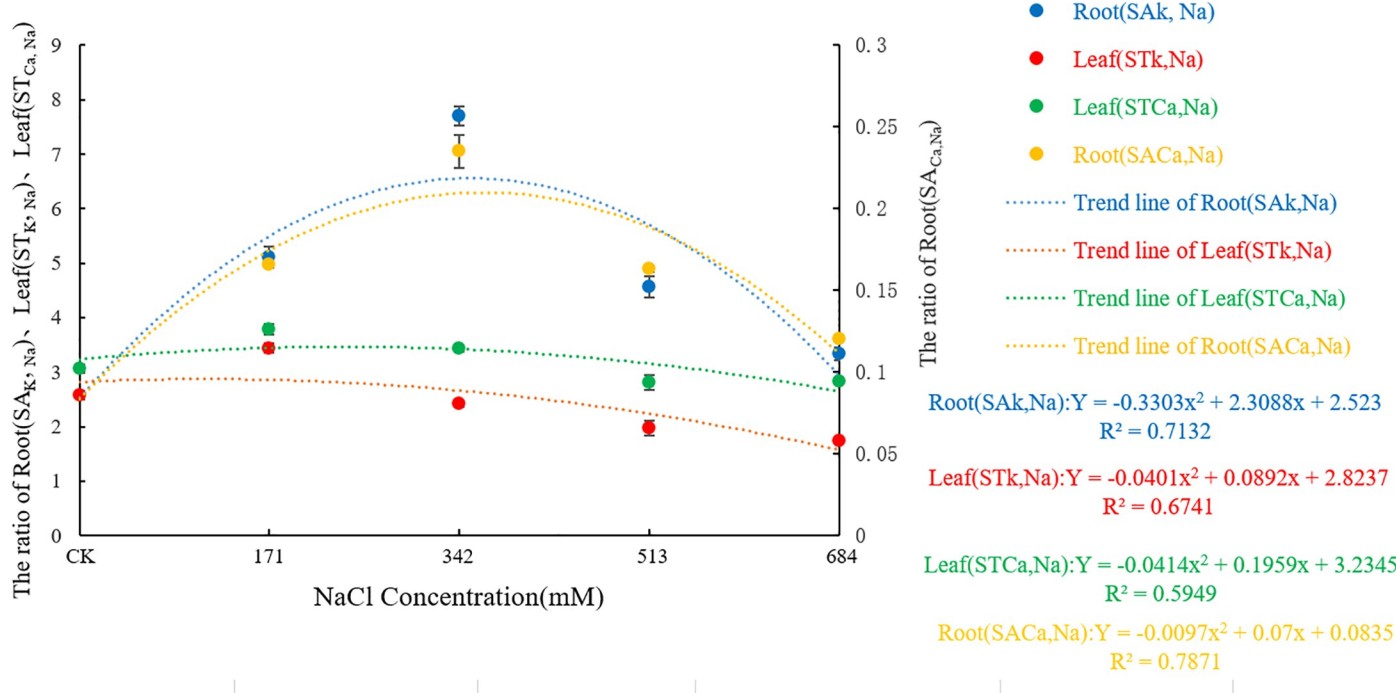

**Fig 7. Effects of salt stress on selective uptake and transportation of ion in roots and leaves of *Salix alba* L.**

significant level (F = 998.922, df = 4, Sig.<0.001;F = 1018.689, df = 4, Sig.<0.001; F = 168.047, df = 4, Sig.<0.001; F = 29.925, df = 4, Sig.<0.001). The selective absorption capacity of roots for $K^+$ is greater than that of $Ca^{2+}$, but the selective transport capacity of the leaves to $Ca^{2+}$ is greater than that of $K^+$. These results indicated that *Salix alba* could adjust the upward transport capacity of $K^+$ and $Ca^{2+}$ via the selective absorption and accumulation of mineral ions to compensate for the change in concentration under salt stress, to prevent the impacts of nutrient deficiency and ion toxicity on the shoot growth.

### 3.3 Effects of salt stress on photosynthetic parameters in *Salix alba* Leaves

Figs 8 and 9 show that the photosynthetic parameters of *Salix alba* leaves were affected to different degrees under different salt concentrations. When the NaCl concentration was 171mM, the $P_n$ of the leaves increased, but there was no significant difference from the control. Later, as the salt stress intensified, the photosynthetic carbon assimilation ability of *Salix alba* leaves was significantly inhibited (F = 95.66, df = 4, Sig.<0.001); when the NaCl concentration was greater than 171mM, both the E (F = 100.091, df = 4, Sig.<0.001)and $g_s$ (F = 69.346, df = 4, Sig.<0.001) were significantly lower than the control and became stronger; but at a low salt concentration (171mM NaCl), there is no significant difference from the control. With the increased salt concentration, the leaf $C_i$ showed a trend of first decreasing and then increasing, reaching the lowest when the salt concentration was 342mM, which was significantly lower than the control by 10.4%, and then it gradually increased. The $C_i$ of leaves (F = 20.50, df = 4, Sig.<0.001) under 513mM and 684mM NaCl treatments were not significantly different from that of the control, but they were significantly higher than the lowest value by 12.4% and 14.6%, respectively.

### 3.4 Effect of salt stress on the rapid chlorophyll fluorescence induction kinetic curve (OJIP) of *Salix alba* leaves

The OJIP curve can provide a great deal of photochemical information about PS and accurately reflect the state of the plant photosynthetic apparatus and the electron redox state of the

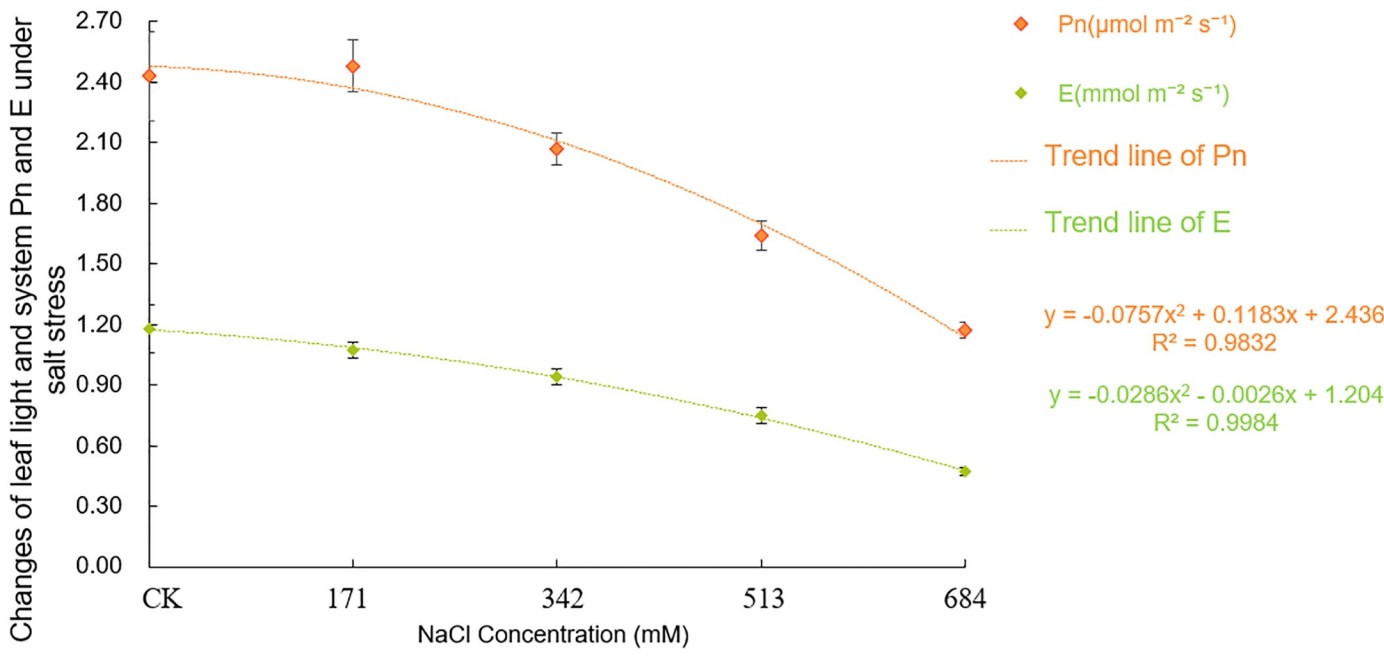

**Fig 8. Effects of salt stress on photosynthetic parameter (Pn and E)in leaves of *Salix alba* L.**

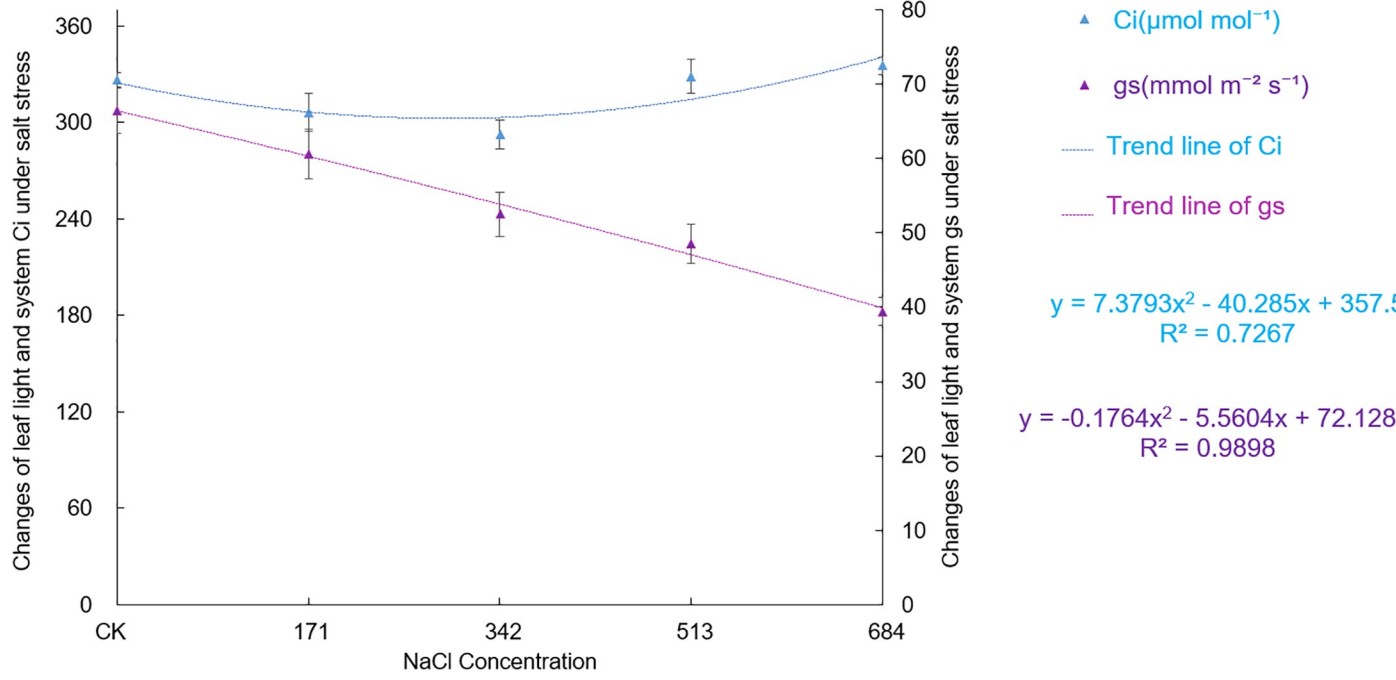

**Fig 9. Effects of salt stress on photosynthetic parameter (Ci and gs) in leaves of *Salix alba* L.**

PS donor side, acceptor side and PS reaction center in the photoreaction [32], thus representing the effects of external stress on the plant photosynthesis ability and even the degree of damage to the photosynthetic organs. Fig 10 shows that with the increasing NaCl concentration, the OJIP curve of *Salix alba* leaves changes to different degrees. Compared with the control group, under the 171mM NaCl treatment, the fluorescence value of JIP does not change significantly; when the NaCl concentration reaches 342mM and higher, the fluorescence values of I and P drop significantly and there is an obvious inflection point K (approximately 300 μs), and the OJIP curve changes to the O-K-J-I-P curve. The K-phase fluorescence value under high salt treatment is higher than that under low salt treatment, and the maximum fluorescence can be reached faster, which indicates that the higher the salt treatment concentration is, the greater the damage to the leaves of *Salix alba*.

### 3.5 Effects of salt stress on quantum yield and energy distribution ratio

Fig 11 shows that under different salt stresses, the energy absorbed, transformed, used for electron transfer, and dissipated by thermal radiation in the leaves of *Salix alba* changes. Compared with the control group, with the increasing NaCl concentration, the maximum photochemical efficiency ($\varphi Po$) of *Salix alba* leaves after dark adaptation gradually decreased. Under the 342mM NaCl treatment, the $\varphi Po$ was significantly lower than that of the control. At that time, salt stress triggered photoinhibition, and the photosynthetic capacity of the leaves was reduced.

The excitons captured by the reaction center transfer electrons to the electron transport chain, and the ratio of excitons that exceed $Q_A$'s other electron acceptors to promote $Q_A$ reduction excitons ($\Psi o$) and the light energy absorbed by the reaction center are used for electron transfer. The quantum yields ($\varphi Eo$) all increased first and then decreased with the increasing salt stress. At 171mM NaCl, although the $\Psi o$ and $\varphi Eo$ increased, they were not significantly different from the control. Later, as the stress intensified, both the $\Psi o$ and $\varphi Eo$

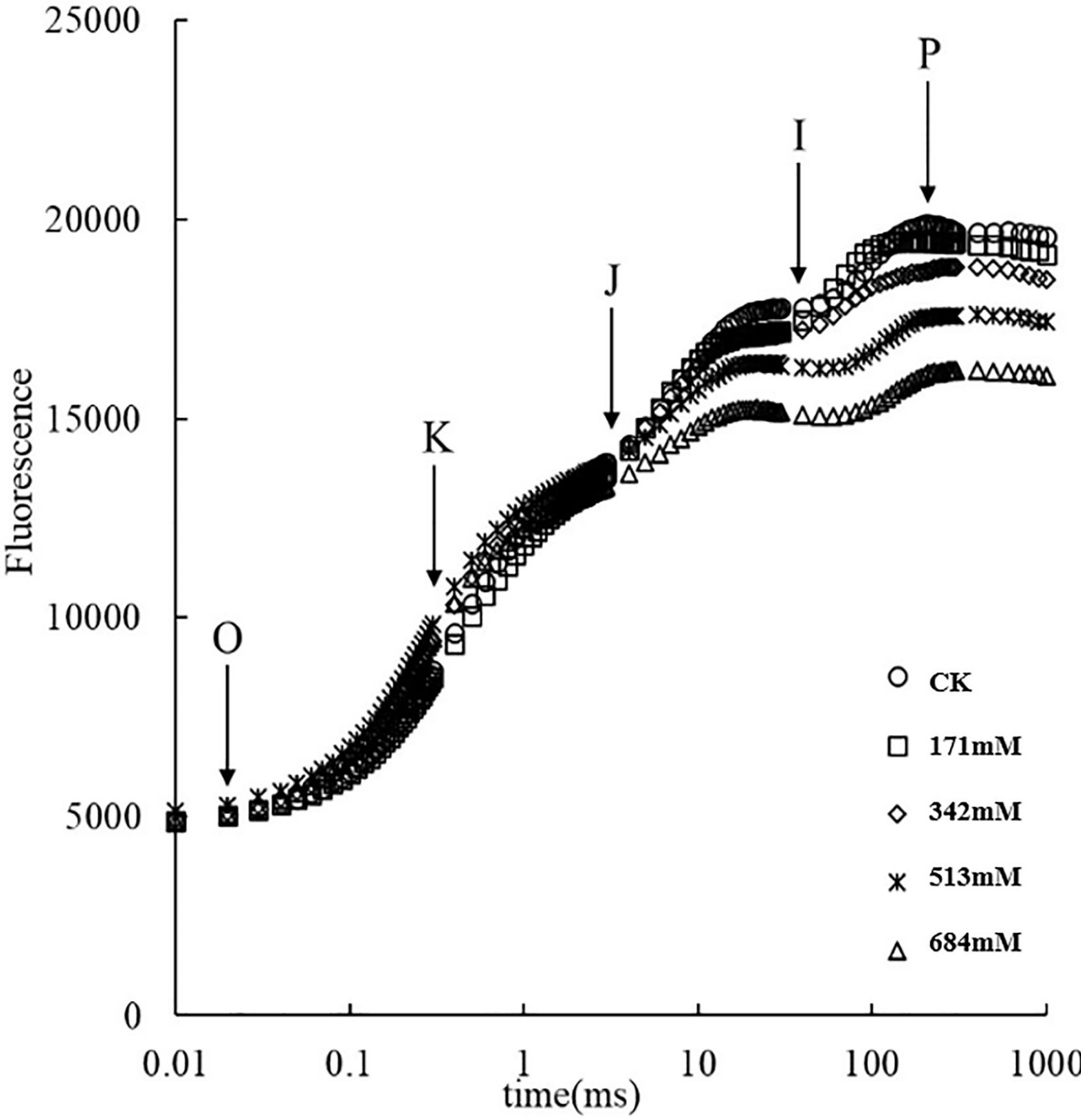

**Fig 10. Effect of salt stress on the fast induction curves of chlorophyll a fluorescence (O-J-I-P curve) of *Salix alba* L. leave.**

were significantly lower than those of the control. When the NaCl concentration was 342mM, the $\Psi$o and $\varphi$Eo were significantly lower than the 11.1% and 11.9% of the control group, respectively. Compared with the control group, salt stress increased the quantum ratio ($\varphi$Do) of *Salix alba* leaves for heat dissipation. When the NaCl concentration was 513mM, $\varphi$Do was significantly higher than that of the control.

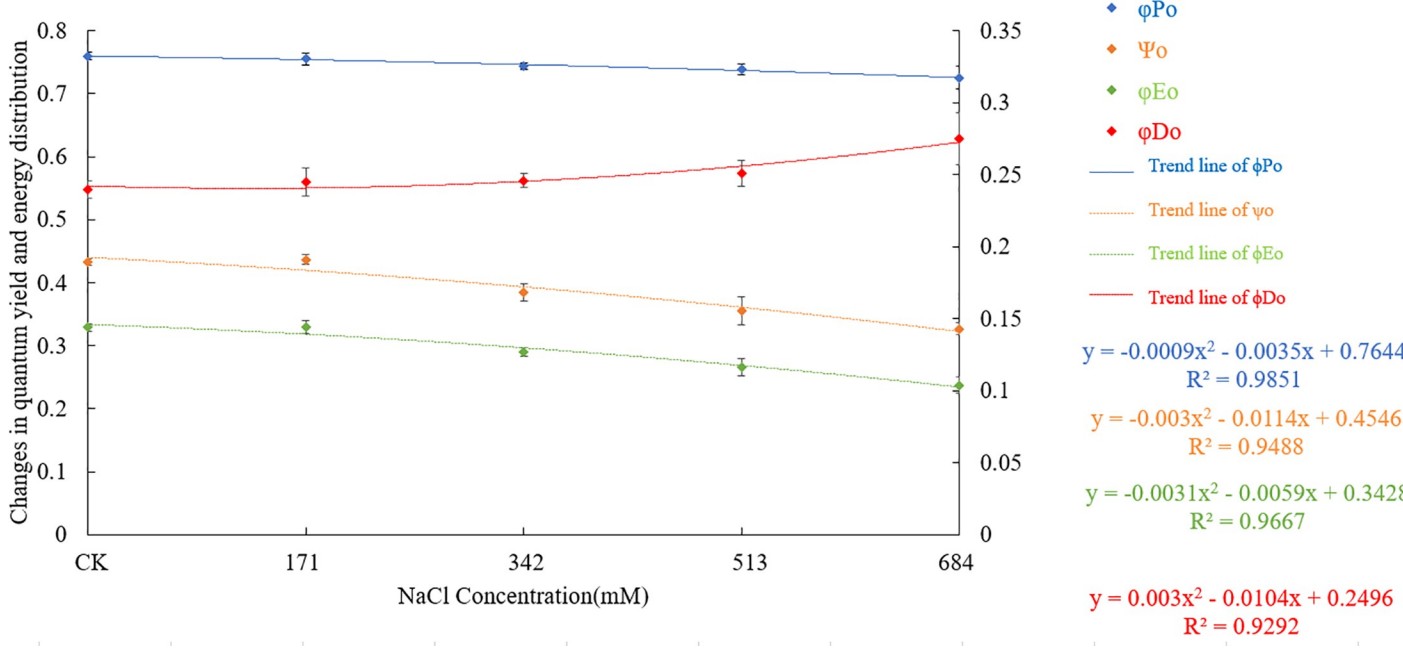

**Fig 11. Changes of chlorophyll a fluorescence parameters under salt stress of *Salix alba* L.**

### 3.6 Influence of salt stress on the performance index and driving force

The performance index and driving force can accurately reflect the changes in the state of the photosynthetic apparatus of plants under stress. $PI_{ABS}$ refers to the performance index based on the absorption of light energy, $PI_{CSm}$ refers to the performance index based on the unit area, and $DF_{CSm}$ refers to the driving force based on the unit area of the material. Figs 12 and 13 show that as the NaCl stress concentration increases, $PI_{ABS}$, $PI_{CSm}$ and $DF_{CSm}$ all show a gradual decline. $PI_{ABS}$ showed no significant difference from the control when the NaCl concentration was 171mM, and then with the increased salt concentrations, the difference became more significant (F = 61.074, df = 4, Sig.<0.001), indicating that the *Salix alba* leaves experienced photoinhibition, the PS was damaged, and the measurement at the 684mM NaCl concentration was significantly lower than that of the control, by 60.2%. When the NaCl concentrations were 342mM, 513mM and 684mM, the $PI_{CSm}$ values were significantly lower (F = 202.821, df = 4, Sig.<0.001) than that of the control by 20.1%, 43.9% and 66.4%; when the NaCl concentrations were 513mM and 684mM, the $DF_{CSm}$ values were significantly lower (F = 40.755, df = 4, Sig.<0.001) than the control by 6.3% and 11.2%. Salt stress seriously affects the absorption of light energy by plants and leads to a decline in the basic driving force.

## 4 Conclusions and discussion

### 4.1 Influence of salt stress on the root growth status of *Salix alba*

As the primary organ responsible for plant material exchange, the root system and its growth status are closely related to the growth and development of the aboveground plant parts, whether the root system can function normally, and the plant's water and nutrient utilization efficiency [33]. Under salt stress, the root system is the first to feel the adversity stress signal, and it is also the most directly affected part [34]. Its ring-stripe inhibition is primarily manifested in the low levels of the root length, surface area and other parameters, and the root

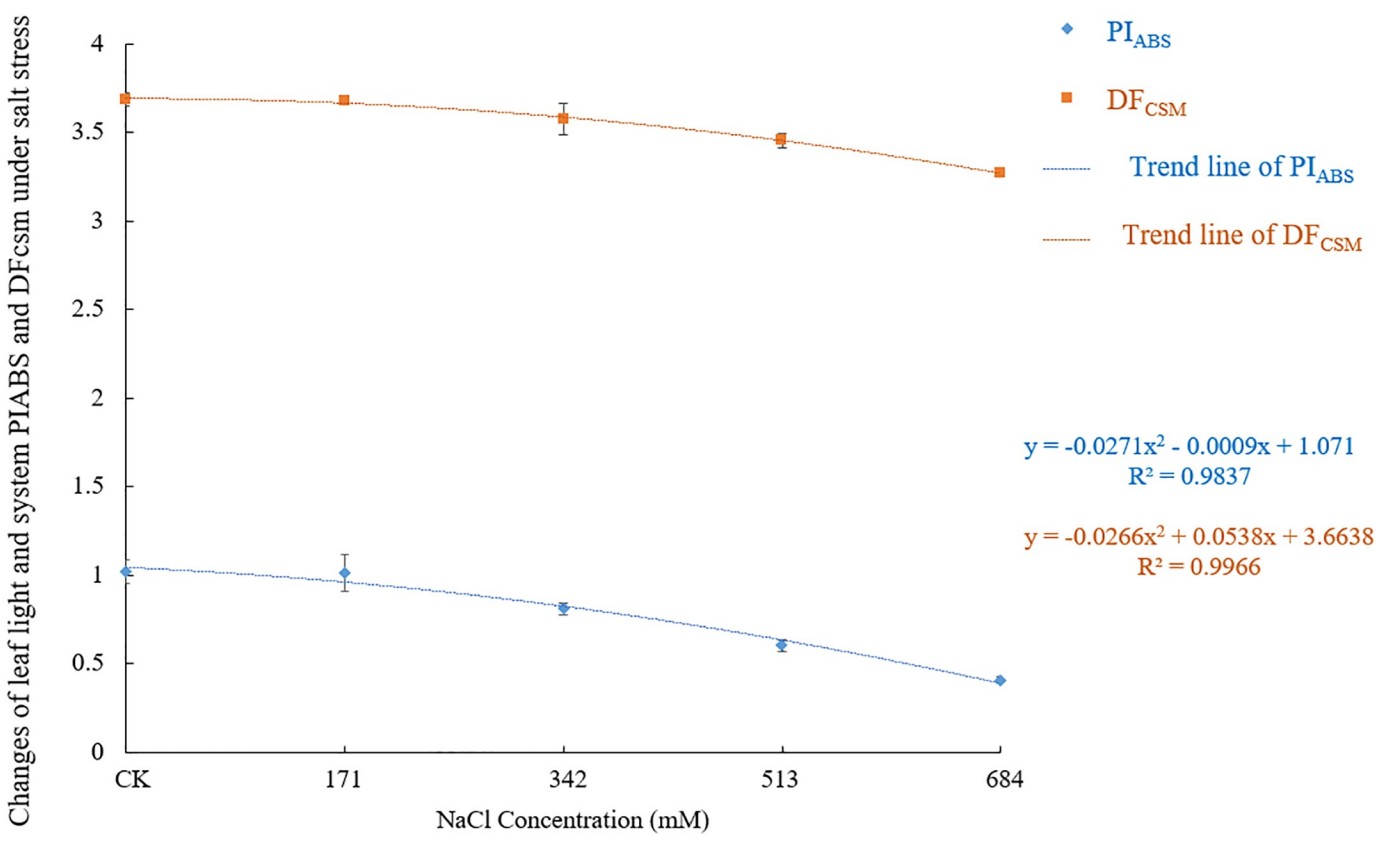

**Fig 12. The changes of performance index and driving force (PI$_{ABS}$ and DF$_{CSm}$) under different salt stress.**

system grows slowly. A high-salt environment will cause plants to experience osmotic stress and ion toxicity, which will lead to changes in membrane permeability, which will in turn affect the absorption of water and nutrient elements by the roots, causing the plants to lose a large amount of water; the ions near the roots will be unbalanced, the physiological functions of the roots will eventually be lowered, and even the structure will be destroyed. Some of the aboveground leaves wilt, and photosynthetic production cannot be performed normally, which causes plant growth and metabolic disorders until the loss of physiological functions.

The change in root growth and the time of the root sprouting period can directly reflect the degree of damage to plants by salt stress and represent the strength of plant salt tolerance [35]. This study showed that the 171mM NaCl concentration significantly promoted the increase in the average number of roots and the elongation of the average root length of *Salix alba* cuttings, and it can promote the rooting of the root system in advance, to a certain extent, which is consistent with Wang Shufeng et al. [36] and Ci Dun. The research results of Wei et al. [35] were basically the same. This growth response may be due to the decrease in water potential outside the roots under salt stress, which stimulates the growth of the roots instead of moderate osmotic stress to ensure the normal absorption of water and nutrients to meet the physiological and metabolic needs of the aboveground parts.

Some plants do have the phenomenon that low salt promotes the increase of some indicators, such as: promoting the germination of sorghum seeds [37], the roots of the seedlings of wolfberry [38] and rice [39], and the growth indicators of corn [40, 41]. Both Chorophyin chrysanthemum [42] and proline content of cherry seedlings [43] are increased, while the net

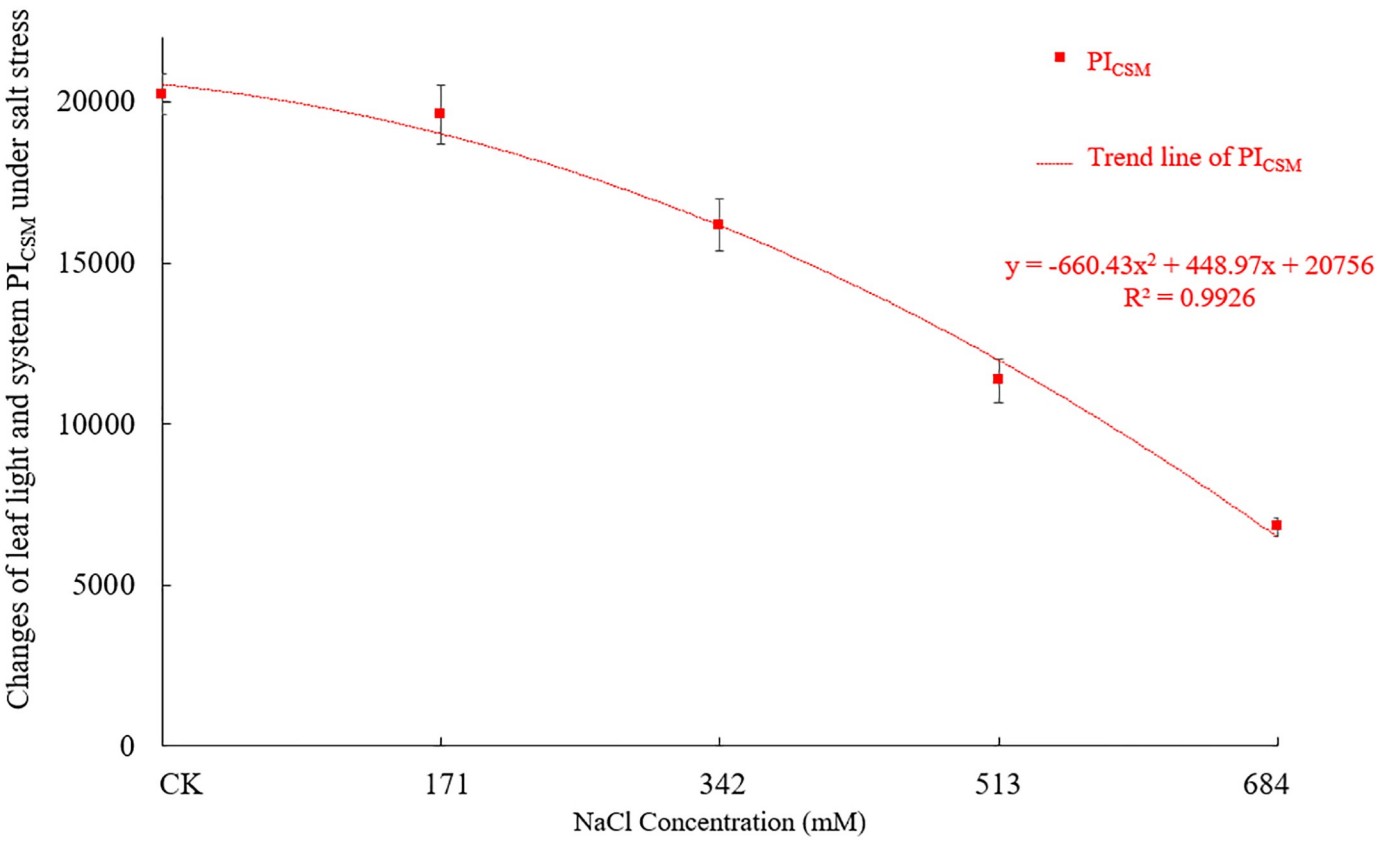

**Fig 13. The changes of performance index and driving force (PI$_{CSm}$) under different salt stress.**

photosynthetic rate of wild chrysanthemum [44] and hazel trees increased [45]. The reason for the low salt concentration may be that the salt stress has a dual effect of stimulus and inhibition on plants. The strong and weak relationship between stimulus and inhibition triggers changes in various plant indicators, resulting in the same low salt. It can promote growth, and it will be inhibited after high salt. This finding shows that *Salix alba* has some ability to adjust and adapt to salt stress, and this adaptability is of great significance to the survival and continuation of the plant itself under adversity. However, as the salt stress intensifies, the ability of plants to coordinate their own growth is destroyed, root germination and elongation are significantly inhibited and become more intense, the root functions are destroyed, and the plants cannot maintain their normal growth and development.

## 4.2 Effects of salt stress on ion content, absorption and transport in *Salix alba*

Ions play an important role in the normal growth of plants, but salt stress can destroy the dynamic balance of ions in plants [46], hinder the absorption of nutrients, and cause plant metabolism disorders. The change in the distribution of ions reflects the degree of damage to plant cells by the external adverse environment. Additionally, plants can maintain balanced nutrition by improving the absorption and transport of ions, which also represents the level of plant resistance to stress. When measuring the ion contents of plant roots and leaves, it is helpful to reveal the salt tolerance or salt damage mechanism of plants.

In this study, when the salt concentration was low, the growth of *Salix alba* was basically normal, the symptoms of salt damage were not significant, and the damage was obvious under severe stress. $Na^+$ accumulates significantly in the roots and leaves of *Salix alba* under salt stress, but the $Na^+$ content in different organs is significantly different, and it is primarily concentrated in the roots. This result shows that the willow root system has a compensation mechanism that can reduce the transportation of salt to aboveground parts by enriching $Na^+$ in the root, thereby effectively reducing or delaying the occurrence of salt damage in the aboveground parts. This conclusion is consistent with the study by Hao Han et al. [47]. When the salt stress is too high, this balance is broken, and growth is blocked.

As an important inorganic solute, $K^+$ is essential for reducing the cell osmotic potential and maintaining the water balance. Generally, plants have an antagonistic effect on the absorption of $Na^+$ and $K^+$ [48], and the competition between the two usually leads to a decrease in the $K^+$ content. The loss of $K^+$ will cause changes in the physical structure of the stomata, frustrating photosynthesis [49]. In addition, $K^+$ participates in the metabolism of various enzymes in plants [50]. As salt stress increases, an excessive loss of $K^+$ will lead to $K^+$ dependent enzymes in *Salix alba* The enzyme activity decreases, which affects the metabolic reactions in which it participates. Therefore, if plants are to grow in a salty environment, the selective absorption of $K^+$ by the root system and the transportation of $K^+$ to the ground are particularly important. This study showed that the $K^+$ content in the roots of *Salix alba* significantly decreased with increasing stress, but the $K^+$ in the leaves could be maintained at a high level at a 342mM NaCl concentration and below and even increased significantly when the NaCl concentration was 171mM, according to Zhou Qi et al. [51] A study on *Carpinus chinensis* also confirmed this result. At this time, the value and increase of $Na^+/K^+$ in the roots of the *Salix alba* were greater than that of the leaves, and the SA $_{k, Na}$ and ST $_{k, Na}$ all increased significantly. Studies have shown that under salt stress, the $Na^+/K^+$ value can represent the degree of salt damage to the plant, and the lower $Na^+/K^+$ value of the leaves can help the plant better maintain its growth and photosynthetic function [52], and the SA $_{k, Na}$ and ST $_{k, Na}$ indicates that the plants can better tolerate salt stress [53]. This result shows that at that time, *Salix alba* could maintain a relatively stable leaf $K^+$ content and the normal progress of photosynthesis by restricting the transportation of $Na^+$ from the root to the leaves, increasing the selective absorption of $K^+$ through the plant roots and the ability to transport $K^+$ to the ground. The accumulation of $Na^+$ causes damage to plants, which may be an important mechanism by which *Salix alba* copes with salt stress. Later, with the increase in salt stress, the $K^+$ in the roots and leaves clearly flowed out. A high concentration of $Na^+$ will replace the $Ca^{2+}$ bound to the membrane system, which will damage the integrity of the membrane structure and membrane function, thereby destroying the ion balance in the plant body and causing a large amount of organic solute extravasation [54]. The establishment of $Ca^{2+}$ homeostasis in the cytoplasm is a key condition for salt adaptation [55]. This experiment showed that as the salt stress intensified, the $Ca^{2+}$ content in the *Salix alba* roots continued to decrease, but it could accumulate in the leaves when the NaCl concentration was $\leq$342mM. The results of Jia Yin et al. [56] were similar; the $Na^+/Ca^{2+}$ value of white *Salix* roots was higher than that of the leaves, and the Sa $_{Ca, Na}$ and ST $_{ca, Na}$ were all significantly increased. This result may be due to the large influx of $Na^+$ into the root system under salt stress, activating $Ca^{2+}$ signal transduction, triggering the sodium elimination system to reduce the damage of $Na^+$, and enhancing the selective absorption of $Ca^{2+}$ in leaves, thereby enhancing the selective transport of $Ca^{2+}$ from root to shoot to maintain the low cell osmotic potential and the stability of the cell membrane. In addition, studies have shown that the increase in intracellular $Ca^{2+}$ contents under salt stress can inhibit the outflow of $K^+$, thereby alleviating the damage of salt stress to plants [57]. Therefore, the upward transportation of $Ca^{2+}$ in the roots of *Salix alba* may be an important mechanism for it to maintain

the balance of $K^+$ and $Na^+$ in the aerial part, establish ion homeostasis in the aerial part, and adapt to salt stress. However, due to the limited ability of the roots of *Salix alba* to absorb $Ca^{2+}$, under high salt stress, the absorption of the roots will not be able to offset the loss of nutrient elements caused by ion poisoning.

## 4.3 Effects of salt stress on photosynthetic parameters of *Salix alba*

Photosynthesis is a key metabolic process that provides material energy for plants. High salt stress will comprehensively affect the photosynthesis of plants through osmotic stress, ion toxicity, and feedback inhibition caused by the accumulation of photosynthetic products [58]. These effects will cause the destruction of the membrane structure and the imbalance of ions in tissue cells, affecting the absorption of light energy by plants and the process of carbon assimilation [59]. This change inhibits the formation of leaf primordia and reduces the photosynthetic area and carbon assimilation of individual plants, resulting in physiological metabolic disorders and the accumulation of toxic substances. In fact, the energy supply related to photosynthesis, carbohydrate metabolism, and the TCA cycle are all inhibited by salt stress [60].

Because stomata are directly connected to the external environment, their coordinated response under stress determines whether the photosynthetic capacity of the plant is normal [61]. In this experiment, the $P_n$, E, and $g_s$ did not change significantly when the NaCl concentration was 171mM. As the salt concentration further increased, each index decreased significantly, which is basically consistent with the results of previous studies [62, 63]. When the NaCl concentration was less than 342mM, the $C_i$ of the *Salix alba* leaves decreased with decreasing $g_s$. Thus, the diffusion resistance of $CO_2$ in the leaves increases, and the carbon sequestration ability weakens. The stoma factor is the dominant factor restricting the decline in *Salix alba* leaf photosynthesis. Later, as the degree of salt stress further intensified, the $C_i$ increased with the decreasing $g_s$, and the photosynthetic system activity of the mesophyll cells decreased, resulting in a decrease in the assimilation capacity, which is a typical non-stomatal limiting factor. Previous studies have shown that under adverse stress, stomatal restriction and non-stomatal restriction and the interaction of the two will reduce the photosynthetic rate of plants; under mild stress, stomatal restriction is dominant; and under severe stress, stomatal restriction leads to non-stomatal restriction [64, 65]. Our experiment also supports this view.

## 4.4 Effects of salt stress on chlorophyll fluorescence kinetics of *Salix alba*

The OJIP curve contains a great deal of information about the original photochemical reaction of the PSII reaction center [66]. When environmental conditions change, chlorophyll fluorescence can directly or indirectly affect the photosystem performance of plants [67]. The changes in the PSII can reflect the impact of changes in the stress environment on the photosynthetic capacity of plants and the adaptation mechanism of photosynthetic machinery to environmental changes. High salt stress can inhibit or destroy parts of the functions of PS, hinder the original photochemical reaction and electron transfer process of PS, and reduce the photosynthetic capacity of *Salix alba* leaves. This consequence may be the result of the accumulation of $Na^+$. The typical fast fluorescence kinetics curve generally has O, J, I, and P phases during the rising phase of fluorescence [68]. This study shows that when the concentration of NaCl is $\geq$ 342mM, the OJIP curve of *Salix alba* will be deformed to OKJIP, the fluorescence values of points I and P will decrease significantly, and obvious inflection point K will appear. The occurrence of the K point is caused by damage to the PSII donor side oxygen release complex (OEC) due to the inhibition of the water lysis system and the receptor-side part before $Q_A$, and the relatively variable fluorescence of the K point can represent the degree of OEC damage [69,

[70]. In addition, the high salt treatment greatly shortened the time required to reach the P point (the maximum fluorescence value). This result indicates that the higher the degree of salt stress, the greater the damage to the stability of the PS reaction center and the OEC on the PS donor side of *Salix alba* leaves, the weaker the ability to provide electrons downstream and the stronger the reduction of the PS acceptor side is hindered.

The φPo, Ψo, φEo, φDo reflect the energy distribution ratio of plants. In this study, when the NaCl concentration was 171mM, there was no significant difference among the indicators. As the stress intensified, the φPo, Ψo and φEo decreased significantly while the φDo increased significantly, which is different from the results of Huang Qinqin et al. [71]. This finding shows that *Salix alba* adjusted the energy distribution ratio of the PSII reaction center under different degrees of stress. This adjustment occurs to increase the quantum ratio used for heat dissipation and reduce the proportion of energy in photochemical reactions, which is an adaptive regulation mechanism of *Salix alba* under salt stress. The decrease in the φPo, Ψo and φEo indicates that the photosynthetic machinery is clearly damaged, the ability to reduce the $Q_B$ and $P_Q$ on the PSII receptor side is diminished, and the electron transfer process is inhibited. Plants are prone to occur or aggravate photoinhibition in adverse environments [69]. In this study, when the concentration of NaCl was greater than 171mM, the $PI_{ABS}$, $PI_{CSm}$ and $DF_{CSm}$ all showed a significant downward trend. This trend shows that *Salix alba* leaves exhibit photoinhibition, the PSII reaction center is reversibly inactivated or irreversibly degraded, the conversion efficiency of light energy is reduced, and the function of the photosynthetic apparatus is impaired, which restricts the normal progress of photosynthesis.

In this study, 171mM NaCl stress had no significant effect on the growth status of the *Salix alba* root system, ion distribution or photosynthetic fluorescence characteristics and even increased these parameters to a certain extent. As the salt treatment concentration gradually increased, the average root number and length, and rooting index decreased significantly; $Na^+$ accumulated in the root system, $K^+$ and $Ca^{2+}$ were significantly lost; the photosynthetic rate decreased significantly, the PS reaction center was partially inactivated, and the donor side OEC and the electron acceptor on the acceptor side were damaged. *Salix alba* can respond to salt stress by intercepting $Na^+$ in the root system, improving the selective absorption of $K^+$ and $Ca^{2+}$ and the ground transportation capacity, and increasing the quantum ratio used for heat dissipation, indicating that *Salix* willow has some tolerance to salt stress environments.

## Supporting information

**S1 Appendix. Effects of salt stress on the contents of three ions in roots and leaves of *Salix alba L.***
(XLSX)

## Acknowledgments

We express sincere gratitude to all the authors involved in this study.

## Author Contributions

**Conceptualization:** Xin Ran, Xiao Wang.

**Data curation:** Xin Ran, Xiao Wang, Xiaokuan Gao, Haiyong Liang, Bingxiang Liu, Xiaoxi Huang.

**Formal analysis:** Xin Ran, Xiao Wang, Bingxiang Liu.

**Funding acquisition:** Haiyong Liang, Bingxiang Liu.

**Methodology:** Xiaokuan Gao, Bingxiang Liu.

**Resources:** Xiao Wang.

**Writing – original draft:** Xiao Wang, Bingxiang Liu.

**Writing – review & editing:** Xiaokuan Gao, Haiyong Liang, Bingxiang Liu.

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
