## [Decision Letter · Decision Letter 0]

8 Sep 2021

PONE-D-21-23173Effects of salt stress on the photosynthetic physiology and mineral ion absorption and distribution in white willow(Salix alba L.)PLOS ONE

Dear Dr. Liu,

Thank you for submitting your manuscript to PLOS ONE. After careful consideration, we feel that it has merit but does not fully meet PLOS ONE’s publication criteria as it currently stands. Therefore, we invite you to submit a revised version of the manuscript that addresses the points raised during the review process.

We look forward to receiving your revised manuscript.

Kind regards,

Mayank Gururani

Academic Editor

PLOS ONE

Journal Requirements:

Reviewers' comments:

Reviewer's Responses to Questions

**Comments to the Author**

1. Is the manuscript technically sound, and do the data support the conclusions?

Reviewer #1: Yes

Reviewer #2: Yes

Reviewer #3: Partly

2. Has the statistical analysis been performed appropriately and rigorously? 

Reviewer #1: Yes

Reviewer #2: Yes

Reviewer #3: No

3. Have the authors made all data underlying the findings in their manuscript fully available?

Reviewer #1: Yes

Reviewer #2: Yes

Reviewer #3: Yes

4. Is the manuscript presented in an intelligible fashion and written in standard English?

Reviewer #1: No

Reviewer #2: Yes

Reviewer #3: Yes

5. Review Comments to the Author

Reviewer #1: The study is carefully concepted and methodological approach is satisfactory explained. The manuscript deals with interesting and important changes in response to salt stress on the photosynthetic physiology and mineral ion absorption and distribution in white willow (Salix alba L.). Results and discussion portions are well written. The authors also draw an accurate picture from the results. However, few parameters are missing. The authors should review and cite some more relevant references, and follow suggestion given in detail below. Also needs a proper revision of English language

Many sentences are very confusing. English language and writing style needs to be improved sufficiently

Do not repeat words of the title in the Keywords.

Please for the first use Salix alba L., then please follow the only Salix alba in the manuscript.

Add economic and other importance of plant in introduction to make it more valuable.

It is better to NaCl concentration in mM instead of %.

Line: 25- 27. Please site the reference for hydroponics.

Objective of the study needs to be refined.

Line: 98. Which instrument (its name model, company) used for the determination of ions?

In my opinion, chlorophyll content (a, b, and total) and carotenoids needs to be measured. It have direct link with salt stress and adaptability.

NaCl and RWC have direct link when talking about stress and adaptability. So, I will suggest to study RWC (relative water content) under salt stress.

The effect of salinity on physiology is obvious in every plant. Therefore, authors should narrate and conclude that how their research is different from others and what is the new or strong points of this manuscript.

References need to be revised. Many articles on salinity response are published in high impact factor journals. So try to cite them, so everyone can access the references as well. Also, try to cite the latest articles.

Reviewer #2: As strong adaptability to environmental stress, Salix alba L. has great potential for use and promotion in the ecological management of coastal saline-alkali soil. To investigate the salt tolerance mechanism of Salix alba L, Ren et al treated one-year-old Salix alba L. cuttings with different NaCl concentrations in a hydroponics system. The growth of roots, the ion contents, and the photosynthetic fluorescence parameters under different salt concentrations were detected to evaluate the salt adaption.

The manuscript was well written and the logic of story was clear. All the data supported the issue which is lower salt treatment can not affect the growth of Salix alba L., but the higher salt concentration will destroy the growth of roots then to the plants. To defense the salt stress, Salix alba L. could restrict the ion transportation to maintain normal ion content in leaf. However, this mechanism seems to be general in many plants, as the authors mentioned in the discussion part. But the differences between this paper and previous work was not reflected. I think the most problem of this work is the lack of innovation. And also, as lack of data, the conclusions can not be rich and for further discussion.

Reviewer #3: The manuscript titled “Effects of salt stress on the photosynthetic physiology and mineral ion absorption and distribution in white willow (Salix alba L.)” one of classic example of tree species response to salt stress. The manuscript is well organized and language easy to follow although there were few grammatical and Syntex errors besides SI units. The manuscript can be improved so many ways before making the technical comments and some of my major concerns are;

1. Adapting a hydroponic system to research salt stress may be the most recent best strategy, but there is no way of knowing how the experimental setting was done. Authors may include a photo of the same in order to make any relevant comments.

2. Similarly for rooting length, kindly provide some photo to understand plant response.

3. Throughout the manuscript I can see that salt stress has a significant impact on above-ground biomass, such as leaf area index, shoot length, and so on, but the fact that it was not included in this study is a significant disadvantage. In the meantime, various physiological characteristics have been subjected leaves.

4. When it comes to hydroponic systems, CK medium may be the best option, however, the reaction of Salix alba seedlings is substantially higher than 0.1% NaCl, but no justification from authors how or why although the salt concentration less than 0.1% NaCl.

5. A statistical analysis was performed, however it was not up to scientific merit. For example, the manuscript frequently mentions significance of attributes, but there are no ANOVA tables (except LSD rank), at least as a supplemental to support. I request the authors to include F=xxx, df=xxx, and sig.=>0.001 wherever statistical significance was indicated.

6. Throughout the article, there are discussions about the trends of the morphological and physiological response of plant to the treatments, but the presented table does not provide viewpoints. Authors profusely mentioned the response trends widely without statistical analysis about trends which containing the scatter plot and best fit regression models (r2 and P). I advise authors to use graphs instead of tables, and I've included a sample presentation for ready reference.

6. PLOS authors have the option to publish the peer review history of their article (what does this mean?). If published, this will include your full peer review and any attached files.

Reviewer #1: **Yes: **Dr. Sunjeet Kumar

Reviewer #2: No

Reviewer #3: **Yes: **Edwinraj Esack https://orcid.org/0000-0003-4264-718X

---

## [Author Response · Author response to Decision Letter 0]

28 Sep 2021

Reply to all comments point-by-point (Manuscript Number: PONE-D-21-23173)

The authors’ replies are in blue.

Editor’s comments:

[Reply] We have modified the manuscript according to the PLOS ONE's style requirements.

[Reply] We will provide the relevant accession numbers or DOIs necessary to access our data. 

[Reply] We create a new iD:https://orcid.org/0000-0002-1699-7648

COMMENTS FOR THE AUTHOR:

Reviewer 1# 

- The study is carefully concepted and methodological approach is satisfactory explained. The manuscript deals with interesting and important changes in response to salt stress on the photosynthetic physiology and mineral ion absorption and distribution in white willow (Salix alba L.). Results and discussion portions are well written. The authors also draw an accurate picture from the results.

[Reply] Thank you very much for your valuable feedback. We have made corrections based on the suggestions. Please see the manuscript.

- However, few parameters are missing. The authors should review and cite some more relevant references, and follow suggestion given in detail below. 

[Reply] We have reviewed and cited more relevant references. Please see line 472-495.

- Also needs a proper revision of English language. Many sentences are very confusing. English language and writing style needs to be improved sufficiently.

[Reply] We have modified the English language appropriately. Thank you very much for your valuable suggestions. We will continue our efforts to improve our English in the future.

- Do not repeat words of the title in the Keywords.

[Reply] We have rewrote Keywords. Please see line 41-43.

- Please for the first use Salix alba L., then please follow the only Salix alba in the manuscript.

[Reply] We have rewrote the name throughout the manuscript. Please look at the example from line 15+18.

- Add economic and other importance of plant in introduction to make it more valuable.

[Reply] We have added economic and other importance of plant in introduction. See Line 64-69 .

- It is better to NaCl concentration in mM instead of %.

[Reply] We've changed % to mM. Please see line 91.

- Line: 25- 27. Please site the reference for hydroponics.

[Reply] We've cited the reference for hydroponics. Please see Line 89 +92-95+441-447.

- Objective of the study needs to be refined.

[Reply] We have refined our research objectives. Please see Line 14-17.

- Line: 98. Which instrument (its name model, company) used for the determination of ions?

[Reply] We have added information about this instrument. We used the atomic absorption spectrometer of Analytikjena in Germany for atomic absorption determination.Please see line 116-117.

- In my opinion, chlorophyll content (a, b, and total) and carotenoids needs to be measured. It have direct link with salt stress and adaptability. NaCl and RWC have direct link when talking about stress and adaptability. So, I will suggest to study RWC (relative water content) under salt stress.

[Reply] This study mainly focuses on the effects of salt ions in plants on photosynthetic performance, without considering chlorophyⅡ and water. Thank you for your valuable suggestions, which will be added and improved in our future studies.

- References need to be revised. Many articles on salinity response are published in high impact factor journals. So try to cite them, so everyone can access the references as well. Also, try to cite the latest articles.

[Reply] We have deleted some references. We have added some influential and recent articles. Please see line 472-495.

COMMENTS FOR THE AUTHOR:

Reviewer 2# 

- However, this mechanism seems to be general in many plants, and also, as lack of data, the conclusions can not be rich and for further discussion.

[Reply] Thank you very much for your suggestions. We have revised the manuscript. Most of the researches on White Willow mainly focus on the medicinal value of substances such as salicin contained in the bark, or the value of studying the enrichment of heavy metals in white willow, which is mainly used to purify water resources and realize agricultural irrigation and fishery breeding. Habitat stress is mainly drought and flooding, but there are few literatures on salt stress, most of which focus on the responses of physiological indexes and photosynthetic indexes of plants to ion absorption and transport under salt stress. This is also one of the reasons for our research. We hope to use white willow as experimental material to observe the various effects of salt stress on plants from this perspective.

COMMENTS FOR THE AUTHOR:

Reviewer 3# 

The manuscript titled “Effects of salt stress on the photosynthetic physiology and mineral ion absorption and distribution in white willow (Salix alba L.)” one of classic example of tree species response to salt stress. The manuscript is well organized and language easy to follow although there were few grammatical and Syntex errors besides SI units.

[Reply] Thank you very much for your suggestions. We have corrected grammatical and syntactic errors .Please see the manuscript.

-1. Adapting a hydroponic system to research salt stress may be the most recent best strategy, but there is no way of knowing how the experimental setting was done. Authors may include a photo of the same in order to make any relevant comments.

[Reply] We have added a picture to the manuscript. Please see the Figure 1.

-2.Similarly for rooting length, kindly provide some photo to understand plant response. 

[Reply] We have added a photo. Please see the figure 2.

-3. Throughout the manuscript I can see that salt stress has a significant impact on above-ground biomass, such as leaf area index, shoot length, and so on, but the fact that it was not included in this study is a significant disadvantage. In the meantime, various physiological characteristics have been subjected leaves.

[Reply] This study mainly focuses on the effects of salt ions in plants on photosynthetic performance, without considering the aspects of leaves and stems. Thank you for your valuable suggestions, which will be added and improved in future research.

-4. When it comes to hydroponic systems, CK medium may be the best option, however, the reaction of Salix alba seedlings is substantially higher than 0.1% NaCl, but no justification from authors how or why although the salt concentration less than 0.1% NaCl.

[Reply] we have made an explanation, please see line 263-269.Thank you very much for your valuable suggestions. 

-5. A statistical analysis was performed, however it was not up to scientific merit. For example, the manuscript frequently mentions significance of attributes, but there are no ANOVA tables (except LSD rank), at least as a supplemental to support. I request the authors to include F=xxx, df=xxx, and sig.=>0.001 wherever statistical significance was indicated.

[Reply] We added some these information throughout the manuscript. Please see the example form line 173-175+179+181.

6.Throughout the article, there are discussions about the trends of the morphological and physiological response of plant to the treatments, but the presented table does not provide viewpoints. Authors profusely mentioned the response trends widely without statistical analysis about trends which containing the scatter plot and best fit regression models (r2 and P). I advise authors to use graphs instead of tables, and I've included a sample presentation for ready reference.

[Reply] Thank you very much for your valuable suggestions. We have changed some tables into charts. Please see figure 7.

---

## [Decision Letter · Decision Letter 1]

11 Oct 2021

PONE-D-21-23173R1Effects of salt stress on the photosynthetic physiology and mineral ion absorption and distribution in white willow(Salix alba L.)PLOS ONE

Dear Dr. Liu,

Thank you for submitting your manuscript to PLOS ONE. After careful consideration, we feel that it has merit but does not fully meet PLOS ONE’s publication criteria as it currently stands. Therefore, we invite you to submit a revised version of the manuscript that addresses the points raised during the review process.

We look forward to receiving your revised manuscript.

Kind regards,

Mayank Gururani

Academic Editor

PLOS ONE

Journal Requirements:

Additional Editor Comments (if provided):

Reviewers' comments:

Reviewer's Responses to Questions

**Comments to the Author**

1. If the authors have adequately addressed your comments raised in a previous round of review and you feel that this manuscript is now acceptable for publication, you may indicate that here to bypass the “Comments to the Author” section, enter your conflict of interest statement in the “Confidential to Editor” section, and submit your "Accept" recommendation.

Reviewer #1: All comments have been addressed

Reviewer #3: (No Response)

2. Is the manuscript technically sound, and do the data support the conclusions?

Reviewer #1: Yes

Reviewer #3: Partly

3. Has the statistical analysis been performed appropriately and rigorously? 

Reviewer #1: Yes

Reviewer #3: No

4. Have the authors made all data underlying the findings in their manuscript fully available?

Reviewer #1: Yes

Reviewer #3: Yes

5. Is the manuscript presented in an intelligible fashion and written in standard English?

Reviewer #1: No

Reviewer #3: No

6. Review Comments to the Author

Reviewer #1: All the comments have been addressed. Few minor corrections and suggestion are given below, which needs to be improve

Proper writing style is “Salix alba L.”, not “Salix alba. L”. Correct it and L. should not be italic (keywords and introduction). Also delete the dot after alba “Salix alba.”

Line 40-41: you can cite the 10.3390/antiox9100940 manuscript for latest update about the Chinese areas affected with salinity. For global scenario cite 10.3389/fpls.2021.660409 manuscript

No need to write mM/L. mM is explaining the unit by itself

Still there are many typing and spelling mistakes. Revise the manuscript thoroughly.

Reviewer #3: The author has improved the revised manuscript in numerous ways, but it still lacks scientific merit, novelty, and grammatical errors, in addition to serious issues with SI units. Despite the authors' efforts, to revise the manuscript failed to address the serious concerns of reviewers regarding the experiment's leaf/above ground biomass to support the research. However, without any reasonable justification, the authors submitted the manuscript with careless revision.

Next is statistics and graphs part. The authors made a few changes, but the manuscript required extensive revision. They modified Fig 7, but similar revisions are required for all graphs except Figs 1, 2, and 10, which must be included in the manuscript. The anova results have been included in 165-175 lines, but they must be given wherever the word significant is used.

Figures are should be modified, XY axis legend and labels have to be edited carefully.

The pdf file enclosed contains a list of all typos, grammatical errors, and suggestions.

7. PLOS authors have the option to publish the peer review history of their article (what does this mean?). If published, this will include your full peer review and any attached files.

Reviewer #1: **Yes: **Dr. Sunjeet Kumar

Reviewer #3: **Yes: **Edwinraj Esack

---

## [Author Response · Author response to Decision Letter 1]

24 Oct 2021

REVIEW COMMENTS TO THE AUTHOR:

Reviewer 1# 

1-All the comments have been addressed. Few minor corrections and suggestion are given below, which needs to be improve.

[Reply] Thank you very much for your valuable suggestions. We have corrected the question you raised. Please look at the following words.

2-Proper writing style is “Salix alba L.”, not “Salix alba. L”. Correct it and L. should not be italic (keywords and introduction). Also delete the dot after alba “Salix alba.”

[Reply] Thank you very much for your advice. We have corrected this writing error in its entirety. We changed ‘Salix alba. L. ’ to ‘ Salix alba L. ’ You can look at line 13 for an example.

We used Salix alba L. at the beginning of the manuscript and Salix alba later.

3-Line 40-41: you can cite the 10.3390/antiox9100940 manuscript for latest update about the Chinese areas affected with salinity. For global scenario cite 10.3389/fpls.2021.660409 manuscript.

[Reply] Thank you for providing us with these two articles, they are very helpful to our article, we have quoted them in line 42-44+421-424.

4-No need to write mM/L. mM is explaining the unit by itself.

[Reply] We have changed mM/L for mM throughout the paper.You can look at line 19 for an example.Thank you very much for your suggestions.

5-Still there are many typing and spelling mistakes. Revise the manuscript thoroughly.

[Reply] Thank you for your advice. We have revised the manuscript thoroughly. We will continue to study English hard in the future.

Finally special thanks to you for your valuable advice. Every suggestion you make will make our manuscript better.

REVIEW COMMENTS TO THE AUTHOR:

Reviewer 3# 

1-The author has improved the revised manuscript in numerous ways, but it still lacks scientific merit, novelty, and grammatical errors, in addition to serious issues with SI units.Despite the authors' efforts, to revise the manuscript failed to address the serious concerns of reviewers regarding the experiment's leaf/above ground biomass to support the research.

[Reply] Thank you very much for your valuable suggestions.We have changed the error according to the SI units. Most of the researches on White Willow mainly focus on the medicinal value of substances such as salicin contained in the bark, or the value of studying the enrichment of heavy metals in white willow, which is mainly used to purify water resources and realize agricultural irrigation and fishery breeding. Habitat stress is mainly drought and flooding, but there are few literatures on salt stress, most of which focus on the responses of physiological indexes and photosynthetic indexes of plants to ion absorption and transport under salt stress. This is also one of the reasons for our research. We hope to use white willow as experimental material to observe the various effects of salt stress on plants from this perspective. 

 In this study, water culture was used to set different salinity gradient, and hard branch cutting was used to analyze the rooting of willow in different salinity. All these measures were to provide theoretical support for willow seedling cultivation in saline-alkali land. In this study, the growth of root system was the key factor to the survival of seedlings, so the rooting index was used in this study. The aboveground biomass can not be used to evaluate the condition of the seedlings because the shoots will die even if they do not take root.

2-Next is statistics and graphs part. The authors made a few changes, but the manuscript required extensive revision. They modified Fig 7, but similar revisions are required for all graphs except Figs 1, 2, and 10, which must be included in the manuscript.

[Reply] Thank you for your advice. We have modified Figs 3,4,5,6,8,9,11,12 and 13. We also used PACE to check our figures.

3-The anova results have been included in 165-175 lines, but they must be given wherever the word significant is used.

[Reply] Thank you very much for your valuable suggestions.We put these data next to the word ‘significant’. Please see line 177-186.

4-Figures are should be modified, XY axis legend and labels have to be edited carefully.

[Reply]Thank you very much for your suggestions. We revised all our Figures according to the journal's requirements.

5-The pdf file enclosed contains a list of all typos, grammatical errors, and suggestions.

[Reply]Thank you very much for your valuable advice. We corrected these errors according to this PDF file. 

We will continue to improve our English in the future. Once again, thank you very much for your valuable suggestions..

We appreciate for Editors and reviewers’ warm work earnestly, and hope that the corrections will meet with approval.

---

## [Editor Report · Decision Letter 2]

1 Nov 2021

PONE-D-21-23173R2Effects of salt stress on the photosynthetic physiology and mineral ion absorption and distribution in white willow(Salix alba L.)PLOS ONE

Dear Dr. Liu,

As you had requested earlier, I am sending this submission back to you so you can upload the correct images as mentioned in your email. Please submit your revised manuscript by 10th Nov. 2021. We look forward to receiving your revised manuscript.

Kind regards,

Mayank Gururani

Academic Editor

PLOS ONE

Journal Requirements:

Additional Editor Comments (if provided):

Dear authors,

As you had requested earlier, I am sending this submission back to you so you can upload the correct images as mentioned in your email.

Kindly submit the revised manuscript asap.

Best regards,

Mayank Gururani

Academic Editor

PLOS ONE
---

## [Author Response · Author response to Decision Letter 2]

2 Nov 2021

Reviewer 1# 

1-All the comments have been addressed. Few minor corrections and suggestion are given below, which needs to be improve.

[Reply] Thank you very much for your valuable suggestions. We have corrected the question you raised. Please look at the following words.

2-Proper writing style is “Salix alba L.”, not “Salix alba. L”. Correct it and L. should not be italic (keywords and introduction). Also delete the dot after alba “Salix alba.”

[Reply] Thank you very much for your advice. We have corrected this writing error in its entirety. We changed ‘Salix alba. L. ’ to ‘ Salix alba L. ’ You can look at line 13 for an example.

We used Salix alba L. at the beginning of the manuscript and Salix alba later.

3-Line 40-41: you can cite the 10.3390/antiox9100940 manuscript for latest update about the Chinese areas affected with salinity. For global scenario cite 10.3389/fpls.2021.660409 manuscript.

[Reply] Thank you for providing us with these two articles, they are very helpful to our article, we have quoted them in line 42-44+421-424.

4-No need to write mM/L. mM is explaining the unit by itself.

[Reply] We have changed mM/L for mM throughout the paper.You can look at line 19 for an example.Thank you very much for your suggestions.

5-Still there are many typing and spelling mistakes. Revise the manuscript thoroughly.

[Reply] Thank you for your advice. We have revised the manuscript thoroughly. We will continue to study English hard in the future.

Finally special thanks to you for your valuable advice. Every suggestion you make will make our manuscript better.

Reviewer 3# 

1-The author has improved the revised manuscript in numerous ways, but it still lacks scientific merit, novelty, and grammatical errors, in addition to serious issues with SI units.Despite the authors' efforts, to revise the manuscript failed to address the serious concerns of reviewers regarding the experiment's leaf/above ground biomass to support the research.

[Reply] Thank you very much for your valuable suggestions.We have changed the error according to the SI units. Most of the researches on White Willow mainly focus on the medicinal value of substances such as salicin contained in the bark, or the value of studying the enrichment of heavy metals in white willow, which is mainly used to purify water resources and realize agricultural irrigation and fishery breeding. Habitat stress is mainly drought and flooding, but there are few literatures on salt stress, most of which focus on the responses of physiological indexes and photosynthetic indexes of plants to ion absorption and transport under salt stress. This is also one of the reasons for our research. We hope to use white willow as experimental material to observe the various effects of salt stress on plants from this perspective. 

 In this study, water culture was used to set different salinity gradient, and hard branch cutting was used to analyze the rooting of willow in different salinity. All these measures were to provide theoretical support for willow seedling cultivation in saline-alkali land. In this study, the growth of root system was the key factor to the survival of seedlings, so the rooting index was used in this study. The aboveground biomass can not be used to evaluate the condition of the seedlings because the shoots will die even if they do not take root.

2-Next is statistics and graphs part. The authors made a few changes, but the manuscript required extensive revision. They modified Fig 7, but similar revisions are required for all graphs except Figs 1, 2, and 10, which must be included in the manuscript.

[Reply] Thank you for your advice. We have modified Figs 3,4,5,6,8,9,11,12 and 13. We also used PACE to check our figures.

3-The anova results have been included in 165-175 lines, but they must be given wherever the word significant is used.

[Reply] Thank you very much for your valuable suggestions.We put these data next to the word ‘significant’. Please see line 177-186.

4-Figures are should be modified, XY axis legend and labels have to be edited carefully.

[Reply]Thank you very much for your suggestions. We revised all our Figures according to the journal's requirements.

5-The pdf file enclosed contains a list of all typos, grammatical errors, and suggestions.

[Reply]Thank you very much for your valuable advice. We corrected these errors according to this PDF file. 

We will continue to improve our English in the future. Once again, thank you very much for your valuable suggestions.

We appreciate for Editors and reviewers’ warm work earnestly, and hope that the corrections will meet with approval.

---

## [Editor Report · Decision Letter 3]

3 Nov 2021

Effects of salt stress on the photosynthetic physiology and mineral ion absorption and distribution in white willow(Salix alba L.)

PONE-D-21-23173R3

Dear Dr. Liu,

We’re pleased to inform you that your manuscript has been judged scientifically suitable for publication and will be formally accepted for publication once it meets all outstanding technical requirements.

Kind regards,

Mayank Gururani

Academic Editor

PLOS ONE
---

## [Editor Report · Acceptance letter]

9 Nov 2021

PONE-D-21-23173R3 

Effects of salt stress on the photosynthetic physiology and mineral ion absorption and distribution in white willow (*Salix alba* L.) 

Dear Dr. Liu:

I'm pleased to inform you that your manuscript has been deemed suitable for publication in PLOS ONE. Congratulations! Your manuscript is now with our production department. 

Kind regards, 

on behalf of

Dr. Mayank Gururani 

Academic Editor

PLOS ONE